# Synthesis and Spectral Identification of Three Schiff Bases with a 2-(Piperazin-1-yl)-*N*-(thiophen-2-yl methylene)ethanamine Moiety Acting as Novel Pancreatic Lipase Inhibitors: Thermal, DFT, Antioxidant, Antibacterial, and Molecular Docking Investigations

**DOI:** 10.3390/molecules25092253

**Published:** 2020-05-11

**Authors:** Ismail Warad, Oraib Ali, Anas Al Ali, Nidal Amin Jaradat, Fatima Hussein, Lubna Abdallah, Nabil Al-Zaqri, Ali Alsalme, Fahad A. Alharthi

**Affiliations:** 1Department of Chemistry and Earth Sciences, Qatar University, P.O. Box 2713, Doha, Qatar; 2Department of Chemistry, Science College, An-Najah National University, P.O. Box 7, Nablus, Palestine; i.kh.warad@gmail.com (O.A.); anas.alali@najah.edu (A.A.A.); 3Department of Pharmacy, Faculty of Medicine and Health Sciences, An-Najah National University, P.O. Box 7, Nablus, Palestine; nidaljaradat@najah.edu (N.A.J.); f.huseen@najah.edu (F.H.); 4Department of Biology and Biotechnology Science College, An-Najah National University, P.O. Box 7, Nablus, Palestine; alubna@najah.edu; 5Department of Chemistry, College of Science, King Saud University, P.O. Box 2455, Riyadh 11451, Saudi Arabia; nalzaqri@ksu.edu.sa (N.A.-Z.); aalsalme@ksu.edu.sa (A.A.); fharthi@ksu.edu.sa (F.A.A.)

**Keywords:** Schiff base, antimicrobial, DFT, pancreatic lipase inhibitors

## Abstract

Three new tetradentate NNNS Schiff bases (**L1**–**L3**) derived from 2-(piperidin-4-yl)ethanamine were prepared in high yields. UV–Visible and FTIR spectroscopy were used to monitor the dehydration reaction between 2-(piperidin-4-yl)ethanamine and the corresponding aldehydes. Structures of the derived Schiff bases were deduced by ^1^H and ^13^C NMR, FTIR, UV–Vis, MS, EA, EDS, and TG-derived physical measurements. DFT/B3LYP theoretical calculations for optimization, TD-DFT, frequency, Molecular Electrostatic Potential (MEP), and highest occupied molecular orbital (HOMO) and lowest unoccupied molecular orbital (LUMO) / were performed for **L2**. The in vitro antimicrobial activities of the three Schiff bases were evaluated against several types of bacteria by disk diffusion test using Gentamicin as the standard antibiotic. Schiff bases revealed good antioxidant activity by the DPPH method, and the IC_50_ values were compared to the Trolox standard. Pancreatic porcine lipase inhibition assay of the synthesized compounds revealed promising activity as compared to the Orlistat reference.

## 1. Introduction 

As synthesized organic polychelate ligands, Schiff bases are unsaturated <C=N- compounds [1,2,3]. Recently, several types of Schiff base ligands with N, O, S, and P atoms were made available for structural analysis and biological applications [4,5,6]. Additionally, Schiff bases possess wide applications in several areas of life sciences as catalysts [6,7,8] and photochromic sensors [9,10]. Schiff bases also possess remarkable biochemical applications, such as antifungal, antiviral, antitumor, and antibacterial activities [11,12,13,14,15,16,17].

Schiff bases are considered to be notable ligands for metal ion coordination complexes due to their ease of synthesis, variability in structural design, and wide their range of applications [11,18,19,20]. These ligands were broadly used as polychelator ligands and have revealed high performance in terms of steric characteristics and electronic soft tuning of their metal complexes [21,22,23,24,25,26]. Chemists design Schiff bases as polydentate ligands and their complexes, and these have served several areas of chemistry [11,17,18,19,20,21,22].

In this work, we focused on the synthetic path and structural determination of three new ligands which contain three nitrogen and one sulfur donor atoms as NNNS Schiff base compounds. These bases were obtained by the condensation of 2-(piperidin-4-yl) ethanamine with 1-(5-chlorothiophen-2-yl) ethanone or thiophene-2-carbaldehyde or 5-bromothiophene-2-carbaldehyde, respectively. The structures of the Schiff base compounds were identified using FTIR, elemental analysis, ^1^H and ^13^C NMR, UV–Vis, TG, EDX, and DFT theoretical calculations. The synthesized materials were examined for their antioxidant, antibacterial, and antilipase activities. The antilipase inhibition effect was successfully explained by the theoretical molecular docking studies.

## 2. Results and Discussion

### 2.1. Synthesis

The desired Schiff bases were prepared by condensation of one mole of 2-(piperidin-4-yl) ethanamine with 1.2 mole of the corresponding aldehydes as: 1-(5-chlorothiophen-2-yl) thenone, thiophene-2-carbaldehyde, and 5-bromothiophene-2-carbaldehyde, which afforded the good yields formation of *N*-(1-(5-chlorothiophen-2-yl)ethylidene)-2-(piperazin-1-yl)ethanamine (**L1**), 2-(piperazin-1-yl)-*N*-(thiophen-2-ylmethylene)-ethanamine (**L2**), and *N*-((5-bromothiophen-2-yl)methylene)-2-(piperazin-1-yl)ethanamine (**L3**), respectively (Scheme 1). The synthesized ligands were found to be soluble in common organic solvents. NMR, MS, UV–Vis, FTIR, EA, EDS, and TG physical measurements were used to analyze the structures of the ligands. Theoretical DFT calculations were performed for **L2** and compared to the corresponding experimental results.

### 2.2. ^1^H and ^13^C NMR Investigation 

Experimental ^1^H NMR and ^13^C NMR analyses of the ligands were performed in CDCl_3,_ as explained in the experimental section. The typical ^1^H-NMR and ^13^C NMR spectra of **L3** are illustrated in Figure 1a,b, respectively. Figure 1a shows a sharp broad signal at δ 1.45 ppm that belongs to an NH piperazine proton; two broad peaks at 2.58 and 2.62 ppm were attributed to the 4CH_2_ of the piperazine ring. Two triplet signals at 2.83 and 3.64 ppm with J_H-H_ = 6.0 Hz were cited to =NCH_2_CH_2_N and =NCH_2_CH_2_N, respectively. The thiophene protons were detected as two doublet signals at *δ* 7.01 and 7.21 ppm with J_H-H_ = 8.0 Hz. The signal of the aldehyde proton at 10.50 ppm disappeared, confirming the formation of azomethineproton -N=CH, which was detected as a singlet at 8.23 ppm.

The ^13^C NMR spectra of **L3** revealed two types of carbons, as shown in Figure 1b: (1) the aliphatic type, attributed to piperazine units with δc 38.8, 50.1, 54.5, and 60.2 ppm; and (2) aromatic carbons as four thiophene singlets at 115.3, 129.0, 130.2, 143.3 ppm; the azomethine carbon -N=CH was recorded at 157.6 ppm.

### 2.3. EDS and Mass Spectroscopy Investigations

The compositions of **L1**–**L3** were determined by EDS analysis, elemental analyses, and MS. The mass spectra of **L1** (Figure 2a) exhibited a molecular ion peak [M]^+^, *m*/*z* at 271.0 (theoretical = 271.2). The results are consistent with the proposed molecular formula of each compound. The EDS analysis of **L3**, shown in Figure 2b, contained C, N, S, and Br; the absence of uncited peaks reflects the purity; the existence of no O atom signal reveals the stability of such compounds against atmospheric O_2_ pressure. 

### 2.4. FTIR and DFTIR Spectral Analysis

FTIR spectroscopy served to monitor the condensation reaction during the ligand’s preparation. The formation of the prepared ligands was spectrally confirmed through C=O/C=N shift and N–H disappearance. The IR of thiophene-2-carbaldehyde and 2-(piperidin-4-yl) ethanamine starting materials were recorded before and after condensation to prepare **L2**, as shown in Figure 3. The stretching vibration of C=O in the carbaldehyde at 1658 cm^−1^ (Figure 3a) was reduced by ~28 cm^−1^ due to the C=N- (1625 cm^−1^) group formation, as shown in Figure 3c. The primary N–H stretching vibration in 2-(piperidin-4-yl) ethanamine at 3340 and 3220 cm^−1^ (Figure 3b) totally disappeared, which supported the full condensation process. 

DFTIR theoretical calculation was performed for free **L2**, as seen in Figure 3d. The theoretical and experimental FTIR spectra revealed an acceptable agreement because the DFT-combinatorial calculation was performed for a free molecule in vacuum; meanwhile, the experimental results in solid state were expected to be lower in chemical shift as compared to the DFT-theoretical calculations [27,28]. 

### 2.5. UV–Vis, TD-DFT/B3LYP Spectral and Frontier Molecular Orbitals Calculations

The electronic absorption behavior of **L1**–**L3** was assessed in ethanol at room temperature. The spectra of the three ligands demonstrated two bands in the 250–310 nm region, which is associated to *n–π** and/or *π–π** electron transfer. The condensation reaction was easily monitored by UV changes before and after the reaction, Figure 4a–c shows the absorbance bands of the starting materials together with the **L3** product. Complete difference in the UV spectroscopy behavior was recorded for **L3** with intense transition band at λ_max_
*=* 302 nm (ε = 4.2 × 10^4^ M^−1^·L^−1^) and a weak band at 260 nm (ε = 1.4 × 10^4^ M^−1^·L^−1^) characterizing the formation of the new Schiff base, **L3**. Time-dependent DFT/B3LYP spectral analysis was also performed for **L3** in ethanol; a major band with λ_max_ = 305 nm was collected, as shown in Figure 4d. An excellent match between the theoretical TD-DFT/B3LYP and the experimental UV-measurement analysis was observed. The slight ~3 nm shift might be due to a solvent effect [27,28].

The HOMO/LUMO energy level calculation is helpful to predict the chemical behavior of the desired materials. Several chemical parameters, such as electrophilicity, hardness, symmetry, chemical potential, quantum chemistry terms, electronegativity, and local reactivity can be evaluated from the HOMO/LUMO energy gap [28,29]. Figure 5 shows the HOMO/LUMO orbital shapes together with their energy levels of **L2** in the gaseous phase. The HOMO response is at −0.19143 a.u., while the LUMO is located at −0.04378 a.u. with a ~0.15 a.u. energy gap. The calculated energy gap value revealed the ease of electron excitation from HOMO to LUMO. The HOMO was found to be a predominant molecular orbital, which is consistent with the overall nature of the tetradentate ligand as a strong electron-donor with a high degree of nucleophilicity. The HOMO–LUMO gap is related to chemical reactivity or kinetic stability, and since both have negative values, they decide the chemical stability of the ligand [29]. 

### 2.6. MEP of ***L2***

The MEP is a useful method to predict the interaction of a studied molecule with its neighboring molecules, and is helpful for evaluating points in molecular activities for biological and electrophilic attacks depending on the polarity. To estimate the electrophilic and nucleophilic sites in the *E*-**L2** ligand, MEP/B3LYP was evaluated, as shown in Figure 6. The electrostatic potentials are illustrated by different colors; red as the most negative, blue as the most positive, and green as zero electrostatic potential. The values of the electrostatic potential in Figure 6 decreased in the order: red > orange > yellow > green > blue.

In the structure of the *E*-**L2** ligand, the maximum negative atoms (red 1Nsp^2^ > orange 2Nsp^3^ ) are the preferred sites for electrophilic attack, and the maximum positive atoms (blue thiophene Hs) are a good site for nucleophilic attack. The yellow S-heterocyclic ring fell between the favor of nucleophilic and electrophilic attack.

### 2.7. Thermal Analysis Investigation

In a 20–700 °C temperature range with heating rate = 10 °C/min and under open atmosphere, the thermographic behaviors of the ligands were determined, and the results are presented in Figure 7.

Figure 7 shows the TG curves of **L1**–**L3. L2** displayed low thermal stability up to 80 °C (Figure 7a), **L3** revealed higher thermal stability up to 115 °C (Figure 7b), and **L1** showed the highest thermal stability up to 210 °C (Figure 7c). All the prepared ligands were totally decomposed to light elemental gases, such as CO_2_, H_2_O, SO_2_, NO_2_, BrOx, and ClOx, in a broad step (~100% weight loss). No residual product or intermediate degradation steps were recorded; the classical one-step thermal decomposition mechanism was suggested. 

### 2.8. Antibacterial Screening

The three synthesized compounds were tested for their antibacterial activity using the disk diffusion method, and the obtained results are illustrated in Figure 8. Gentamicin was used as the standard antibiotic. It was observed that the examined compounds were not powerful antibacterial agents. The desired compounds were found to be antibacterial in the following order: **L2** > **L1** > **L3**. In spite of its low antibacterial behavior, **L2** was active against all examined bacterial isolates. The most sensitive isolates to **L2** were Gram-positive *S. aureus* and methicillin-resistant *Staphylococcus aureus* (MRSA), with inhibition zone diameters (IZDs) of 12 mm and 11 mm, respectively. In addition, the growth of *P. aeruginosa* was only inhibited by **L2** (9 mm).

It was found that **L3** had no antibacterial activity against all tested Gram-negative bacteria and it was only active against Gram-positive *S. aureus*, resulting in an 11 mm inhibition zone. Several studies have shown that the in vitro biological evaluation of free compounds against various pathogenic bacterial strains are less powerful than their complexes with metals, such as copper, nickel, zinc, and cobalt [30,31].

### 2.9. Antioxidant Activity 

The three synthesized Schiff bases were evaluated against DPPH radical scavenging antioxidant activity, as seen in Figure 9. IC_50_ values were calculated and compared with those of the Trolox reference. Compound **L1** exhibited higher activity (5.15 µg/mL) than **L3** (10.59 µg/mL) and **L2** (12.58 µg/mL), while Trolox revealed the highest antioxidant activity with IC_50_ = 1.93 µg/mL. The reason why these Schiff bases have antioxidant potential can be explained by the presence of halides in their structures.

In general, highly antioxidant materials should be associated with high antibacterial activities. No clear relations between the antioxidant and antibacterial behaviors of the three investigated Schiff base ligands were recorded. For example, **L1** showed the highest antioxidant activity, but moderate antibacterial activity. 

### 2.10. Antilipase Activity of ***L1***–***L3*** and Their Molecular Docking Investigations

Obesity is a risk factor for many diseases, such as cardiac diseases, cancer, and diabetes mellitus [29]. The lipolytic pancreatic lipase enzyme is synthesized and secreted by the pancreas, which plays a key role in the efficient digestion of the lipids and is responsible for the hydrolysis of 50–70% of total dietary lipids. Antilipase activity is one of the most widely studied mechanisms for the determination of the potential efficacy of newly investigated molecules as antiobesity agents [32,33].

The ability of the prepared Schiff bases to inhibit porcine pancreatic lipase was evaluated using Orlistat as the reference, as seen in Figure 10. **L3** showed the best antilipase activity with IC_50_ = 158.48 µg/mL, followed by **L2** with IC_50_ = 501.18 µg/mL, and **L1** with IC_50_ = 316.22 µg/mL, compared to the standard Orlistat reference with IC_50_ = 120.02 µg/mL.

According to the obtained results, the newly synthesized compound **L3** can be considered a potential inhibitor of pancreatic lipase enzyme and a new player in obesity treatment. In fact, this molecule can be prepared in pharmaceutical formulations to treat or prevent obesity, control overweight, or treat hypertriglyceridemia. Further pharmacological and toxicological in vivo studies are required to prove its therapeutic effects and to prepare suitable pharmaceutical forms.

The molecular docking of the three prepared ligands with pancreatic porcine lipase were performed to explain the experimental measurements. Several significant H-bond interactions were detected between the pancreatic porcine lipase and the docked ligands. The best bonding site for each ligand was chosen based on the lowest binding energy and RMSD [34]. Comparison of the docked sites of the three ligands reflected several loading positions. Moreover, each ligand is surrounded by several different amino acids; for example, **L1** is surrounded by ASP-2, TRP-339, GLU392, and VAL-39, but forms only one H-bond between the N–H….O=C of ASP-2 with 1.819 Å and −6.1 kcal/mol binding energy. Similarly, **L2** was surrounded by GLU-364, ASP-890, GLU-392, and ASP-279, forming its H-bond with N–H…O=C of ASP-279 with 1.830 Å and −6.4 kcal/mol binding energy. Meanwhile, **L3** was surrounded with more amino acids, including TYR-341, GLU-364, CLU-392, ASN-363, ARG-423, ASN-424, and TRP-339, and formed two H-bonds; one with N–H…O=C of GLU-364 with 1.716 Å and the second with –Br…H of ASN-363 with 2.032 Å and −7.0 kcal/mol binding energy. The number and strength of hydrogen bonds formed between ligands and the lipase revealed **L3** as the better binder compared to the other ligands, which is consistent with the experimental lipase inhibition result [34,35,36,37]. Comparing the biological activity of the three studied ligands with similar Schiff bases, the antibacterial and antioxidant activities were very close [31]. In general, Schiff bases have weak antibacterial activity compared to their transition metal ion complexes [26,30,38,39,40,41]. The difference is caused by the presence of active functional groups in the starting materials used to prepare the Schiff bases [30,31,32,33,35]. Therefore, if such compounds are required to have antibacterial activity, the functional groups of amines and carbonyls should be carefully selected. The ability of the synthesized Schiff bases to inhibit porcine pancreatic lipase better than similar ligands can be attributed to the presence of a thiophene functional group in the backbones of the synthesized ligands. 

## 3. Materials and Methods

### 3.1. Material and Instrumentation 

2-(Piperidin-4-yl)ethanamine, 1-(5-chlorothiophen-2-yl)ethanone, thiophene-2-carbaldehyde, 5-bromothiophene-2-carbaldehyde, Trolox, DPPH, p-nitrophenol, Orlistat, porcine pancreatic lipase, and all the solvents used in the study were purchased from Sigma (Missouri, St. Louis, MO, USA) and were used as received. A Perkin-Elmer Spectrum 1000 FTIR Spectrophotometer (PerkinElmer Inc., Waltham, MA, USA) was used for IR. TG/DTG curves of TGA-7 were recorded by Perkin-Elmer. An Elementar-Vario EL analyzer (PerkinElmer Inc., Waltham, MA, USA) performed the elemental analyses. A TU-1901-double-beam UV–Visible spectrophotometer (Purkinje General Instrument Co., Ltd., Beijing, China) was used for UV–Vis spectroscopy. 

### 3.2. Synthesis of Ligands ***L1***–***L3***

A concentration of 0.01 mol of 2-(piperidin-4-yl) ethanamine in 10 mL of dichloromethane was added to 0.012 mol of 1-(5-chlorothiophen-2-yl) ethanone, thiophene-2-carbaldehyde, and 5-bromothiophene-2-carbaldehyde dissolved in 10 mL dichloromethane. The mixture was stirred for 3 h at room temperature. The organic layer that contained the product was extracted with 50 mL/50 mL dichloromethane/distilled water; MgSO_4_ was used as a drying agent to remove any water from the dichloromethane solution. A colorless, viscous, oily product (with 80–90% yield) was collected after evaporation of the dichloromethane solvent. 

### 3.3. N-(1-(5-Chlorothiophen-2-yl)ethylidene)-2-(piperazin-1-yl)ethanamine (***L1***)

The parameters for **L1** are as follows: Yield: 86%, IR main functional groups: 3010 cm^−1^
_C–H thiophene_, 2885 cm^−1^
_C-H aliphatic_**,** 1630 cm^−1^
_C=N_. UV–Vis in EtOH: 288 nm. ^1^H NMR (250 MHz, CDCl_3_): (ppm) 1.32 (s, 3H, CH_3_), 1.45 (br, 1H, NH-piperazine), 2.54, 2.66 (2br, 8H, 4CH_2_-piperazine), 2.92 (t, J_H-H_ = 5.8 Hz, 2H, =NCH_2_CH_2_N), 3.70 (t, J_H-H_ = 6.2 Hz, 2H, =NCH_2_CH_2_N), 7.12, 7.32 (2d, 2H, J_H-H_ = 8.4 Hz, thiophene). ^13^C NMR: (ppm), 16.8 (s, 1C, CH_3_), 40.2 (s, 2C, CH_2_NHCH_2_-piperazine), 52.4 (s, 1C, =NCH_2_CH_2_N), 55.2 (s, 2C, CH_2_NCH_2_-piperazine), 62.2 (s, 1C, =NCH_2_CH_2_N), 120.8, 128.7, 132.8, 145.5 (4s, 4C, thiophene), 162.5 (s, 1C, >C=N-). MS: *m*/*z* 271.8 (M^+^ or C_12_H_18_ClN_3_S^+^). *Anal*. Calcd. For C_12_H_18_ClN_3_S: C, 53.03; H, 6.67; N, 15.46. Found: C, 53.08; H, 6.54; N, 15.28.

### 3.4. 2-(Piperazin-1-yl)-N-(thiophen-2-ylmethylene)ethanamine (***L2***)

The parameters for **L2** are as follows: Yield: 84%, IR main functional groups: 3005 cm^−1^
_C–H thiophene_, 2860 cm^−1^
_C-H aliphatic_**,** 1625 cm^−1^
_C=N_. UV–Vis in EtOH: 290 nm (sharp) and 270 nm shoulder. ^1^H NMR (250 MHz, CDCl_3_): (ppm), 1.50 (s, 1H, HN), 2.50, 2.58 (2br, 8H, 4CH_2_-piperazine), 2.81 (t, J_H-H_ = 6.1 Hz, 2H, =NCH_2_CH_2_N), 3.62 (t, J_H-H_ = 6.2 Hz, 2H, =NCH_2_CH_2_N), 7.12, 7.32 (2d, 2H, J_H-H_ = 7.8 Hz, thiophene), 7.41 (t, 1H, J_H-H_ = 5.2 Hz, thiophene), 8.21 (s, 1H, –HC=N-). ^13^C NMR: (ppm), 37.5 (s, 2C, CH_2_NHCH_2_-piperazine), 50.4 (s, 1C, =NCH_2_CH_2_N), 54.1 (s, 2C, CH_2_NCH_2_- piperazine), 60.0 (s, 1C, =NCH_2_CH_2_N), 115.1, 128.8, 130.0, 143.2 (4s, 4C, thiophene), 158.6 (s, 1C, –HC=N-). MS: *m*/*z* 223.1 (M^+^ or C_11_H_17_N_3_S^+^). *Anal*. Calcd. For C_11_H_17_N_3_S: C, 59.16; H, 7.67; N, 18.81. Found: C, 59.07; H, 7.60; N, 18.70.

### 3.5. N-((5-Bromothiophen-2-yl)methylene)-2-(piperazin-1-yl)ethanamine (***L3***)

The parameters for L3 are as follows: Yield: 90%, IR main functional groups: 3005 cm^−1^ C–H thiophene, 2890 cm^−1^ C-H aliphatic**,** 1628 cm^−1^
_C=N_. UV–Vis in EtOH: λ_max_ at 260 nm (ε = 1.4 × 10^4^ M^−1^·L^−1^) and 302 nm (ε = 4.2 × 10^4^ M^−1^·L^−1^). ^1^H NMR (250 MHz, CDCl_3_): (ppm), 1.45 (br, 1H, NH- piperazine), 2.58, 2.62 (2br, 8H, 4CH_2_- piperazine), 2.83 (t, J_H-H_ = 6.0 Hz, 2H, =NCH_2_CH_2_N), 3.64 (t, J_H-H_ = 6.0 Hz, 2H, =NCH_2_CH_2_N), 7.01, 7.21 (2d, 2H, J_H-H_ = 8.0 Hz, thiophene), 8.23 (s, 1H, –HC=N-).^13^C NMR: (ppm), 38.8 (s, 2C, CH_2_NHCH_2_- piperazine), 50.1 (s, 1C, =NCH_2_CH_2_N), 54.5 (s, 2C, CH_2_NCH_2_- piperazine), 60.2 (s, 1C, =NCH_2_CH_2_N), 115.3, 129.0, 130.2, 143.3 (4s, 4C, thiophene), 157.6 (s, 1C, –HC=N-).MS: *m*/*z* 302.7 (M^+^ or C_11_H_16_BrN_3_S^+^). *Anal*. Calcd. For C_11_H_16_BrN_3_S: C, 43.71; H, 5.34; N, 13.90; Br, 26.44. Found: C, 43.27; H, 5.22; N, 13.78.

### 3.6. Computational

The structure of the prepared ligand was solved by the X-ray diffraction method and the structure was optimized at DFT/B3LYP by the GAUSSIAN09 (09, Gaussian Inc.: Wallingford, CT, USA) [38]. Molecular docking was performed with AutoDock v4.2 (AUTODOCK GmbH, Hamburg, Germany). The lipase molecule was processed by adding all hydrogens and merging nonpolar hydrogen atoms using AutoDock Tools. The charges were assigned using the Gasteiger method [39]. 

### 3.7. Antibacterial Activity 

The antibacterial activity of the prepared ligands was determined by the disk diffusion method (NCCLS, 1999). Their biological activity was evaluated against three reference bacterial isolates: *Staphylococcus aureus* (ATCC25923), *Escherichia coli* (ATCC 25922), and *Pseudomonas aeruginosa* (ATCC 27853). In addition, two clinical isolates, Methicillin-resistant *Staphylococcus aureus* (MRSA) and *Klebsiella pneumonia* were obtained from Rafidia hospital and were identified by the Biology and Biotechnology Laboratory at the An-Najah National University in Nablus, Palestine. The tested bacteria were grown over night on Muller Hinton agar plates (Sigma-Aldrich, Missouri, MO, USA) manufacture, city, state abbreviation if US, country). Broth turbidity was adjusted to 0.5 McFarland (1.5 × 10^8^ CFU). Then, each bacterial culture was inoculated by streaking the swab over the entire sterile agar surface. This procedure was repeated by streaking two more times, rotating the plate approximately 60 °C each time to ensure an even distribution of the inoculums. As the final step, the rim of the agar was also swabbed. After 10 min, 1 mg/mL of each ligand under study was loaded onto 6 mm disks, and then the prepared disks were added to the surface of inoculated agar plates. The plates were allowed to stand at room temperature for 30 min for the ligands to diffuse into the agar and then they were incubated at 37 °C for 18 h. After incubation, all the plates were examined for bacterial growth inhibition by measuring the inhibition zone diameter (IZD) to the nearest mm. The test was performed in duplicate. Gentamicin (G) was used as the positive control.

### 3.8. Antioxidant Activity

Firstly, a stock solution at a concentration of 1 mg/mL in methanol was prepared for each sample and the Trolox reference. The working solutions of the following concentrations (1, 2, 3, 5, 7, 10, 20, 30, 40, 50, 80, 100 μg/mL) were prepared by serial dilution with methanol from the original stock solution. DPPH was freshly prepared at a concentration of 0.002% *w*/*v*. The DPPH solution was mixed with methanol and the above-prepared working concentration in a ratio of 1:1:1, respectively. The spectrophotometer was zeroed using methanol as a blank solution. The first solution of the series concentration was DPPH with methanol only. The solutions were incubated in the dark for 30 min at room temperature; further, the absorbance readings were recorded at 517 nm. The percentage of the antioxidant activity of each sample and the Trolox standard were calculated using the following formula: Percentage of inhibition of DPPH activity (%) = (A − B)/A × 100(1)
where A = optical density of the blank and B = optical density of the sample. 

The antioxidant half maximal inhibitory concentration (IC_50_) for each sample and the standard were calculated using BioDataFit Edition 1.02 software (SigmaPlot, San Jose, CA, USA).

### 3.9. Pancreatic Lipase-Enzyme Inhibition 

Preparation of lipase stock solution and dilution series for the samples: The porcine pancreatic lipase inhibitory assay was adapted from [41] with some modifications. Briefly, a sample stock solution of 1 mg/mL was prepared in 10% DMSO from which five different dilutions were prepared with the concentrations 200, 400, 600, 800, and 1000 μg/mL. The stock solution with 1 mg mL of pancreatic lipase enzyme was prepared immediately before use and was suspended in 10% DMSO solution. The antilipase substrate stock solution of *p*-nitrophenyl butyrate (PNPB) was prepared by dissolving 20.9 mg of PNPB in 2 mL of acetonitrile. For each working test tube, 0.1 mL of porcine pancreatic lipase (1 mg/mL) was added to 0.2 mL of **L1**–**L3** from each diluted solution series. The mixture volume was made up to 1 mL by adding Tri-HCL solution, followed by incubation at 37 °C for 20 min. PNPB solution, 0.1 mL, was added to each test tube. The mixture was incubated for 30 min at 37 °C. Pancreatic lipase activity was examined by evaluating the hydrolysis of *p*-nitrophenolate to *p*-nitrophenol by measuring absorbance at 405 nm using a spectrophotometer. The same procedure was carried out for the Orlistat reference.

## 4. Conclusions

Three new NNNS-Schiff bases, **L1**–**L3,** were synthesized by condensation of 2-(piperidin-4-yl) ethanamine with the corresponding carbonyl compounds. The reaction during the synthesis was successfully monitored by FTIR and UV–Vis spectroscopy. The structures of the prepared ligands were spectrally and theoretically characterized. The DFT/B3LYP theoretical calculations strongly matched their relative experimental results. The in vitro antimicrobial activities of the three Schiff bases were evaluated against several types of bacteria; the result reflected a promising antibacterial effect, and **L2** was found to be the best candidate. The tested Schiff bases exhibited good antioxidant activity. Pancreatic porcine lipase inhibition assays of the synthesized ligands revealed promising antiobesity activity, and the Schiff bases could be further investigated as new antilipase drugs. The molecular docking analysis successfully explained this activity, showing that **L3** formed two short H-bonds with the lipase, and **L1** and **L2** each formed one H-bond.

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
