# Peer review of "Synthesis and Spectral Identification of Three Schiff Bases with a 2-(Piperazin-1-yl)-N-(thiophen-2-yl methylene)ethanamine Moiety Acting as Novel Pancreatic Lipase Inhibitors: Thermal, DFT, Antioxidant, Antibacterial, and Molecular Docking Investigations"

_molecules, 2020, doi:10.3390/molecules25092253_

Round 1

Reviewer 1 Report

This submission reports on the synthesis and chemical/biological characterization of three simple Schiff base ligands that are not really synthetically-challenging to prepare.  The following comments are presented to the authors solely in a constructive manner.  While the paper is well-written, the study is not fully justified (i.e. why were these particular aldehydes and amine chosen for this study and why do the authors think these imines would show any enhanced activities over many of the known derivatives?).  Likewise, their results are not compared well to the known and vast Schiff base ligand literature.  Indeed, this submission just reports on the results of the characterization and biological studies.  The DFT studies are routine and have been done before on related imines.  There are far too many routine figures included in the submission so that it reads much more like a scientific report than a research paper.  Several examples of incorrect use of upper and lower case letters are presented throughout the paper that need to be corrected.  For instance in the experimental section, 2-(piperidin-4-yl)ethanamine is the first word in a sentence so it should be 2-(Piperidin-4-yl)ethanamine as the two is just a designation and P is the first letter in the sentence.  Another example is p-Nitrophenol in the middle of a sentence and should be p-nitrophenol.  As mentioned previously, the results are not compared to what is known in the literature so it is difficult for the reader to understand where this study fits in the wider scheme of things.  There is far too much routine information that could be reduced to a few sentences (and this work should only have about 3 figures).  Finally, this work should be much more concisely presented.  I would therefore recommend publication in Molecules only after minor revision.

Author Response

Thanks for your review

Your comments enriched the manuscript.

This submission reports on the synthesis and chemical/biological characterization of three simple Schiff base ligands that are not really synthetically-challenging to prepare.  The following comments are presented to the authors solely in a constructive manner.  While the paper is well-written, the study is not fully justified (i.e. why were these particular aldehydes and amine chosen for this study and why do the authors think these imines would show any enhanced activities over many of the known derivatives?).  Likewise, their results are not compared well to the known and vast Schiff base ligand literature.  Indeed, this submission just reports on the results of the characterization and biological studies. 

reply 

  • We prepared these Schiff base ligands via coupling of 2-(Piperidin-4-yl)ethanamine with carbonyl compounds since the main nucleus compound is 2-(Piperidin-4-yl)ethanamine, so it expected to be with good biological active, commercial aldehyde and ketone were used to prepare new ligands  (not prepared before). Therefore, such ligands are desires to be new (not prepared before), the aldehydes and ketone are commercially available, such ligands are stable in different media and they will be of biological activity.

The DFT studies are routine and have been done before on related imines.  There are far too many routine figures included in the submission so that it reads much more like a scientific report than a research paper. 

  • I do not mind transferring or deleting any of the figures to Supplementary files and I will leave this option to the journal’s administration in order to avoid any conflicts. Personally, I like to keep it as it’s.

Several examples of incorrect use of upper and lower case letters are presented throughout the paper that need to be corrected.  For instance in the experimental section, 2-(piperidin-4-yl)ethanamine is the first word in a sentence so it should be 2-(Piperidin-4-yl)ethanamine as the two is just a designation and P is the first letter in the sentence.  Another example is p-Nitrophenol in the middle of a sentence and should be p-nitrophenol. 

  • Corrected and highlighted to the text

As mentioned previously, the results are not compared to what is known in the literature so it is difficult for the reader to understand where this study fits in the wider scheme of things. 

At the end of the biological activity, we compared the biological result with a similar system as the following:

  • Comparing the biological activity of the desired three ligands with similar Schiff bases, the antibacterial and antioxidant activities are very close to each 34. In general, Schiff bases are weak antibacterial activity compared to their transition metal ions complexes 26-28, 33, 41. What makes the difference is the presence of originally active functional groups in the starting materials used to prepare the Schiff bases 34. Therefore, if such compounds are required to be antibacterial active the functional groups in amines and carbonyls should be carefully selected. The high ability of the desired Schiff bases to inhibit porcine pancreatic lipase better than similar ligands can be contributed to the presence of thiophene functional group in the backbones of the desired ligands.  

There is far too much routine information that could be reduced to a few sentences (and this work should only have about 3 figures).  

  • I do not mind transferring or deleting any of the figures to Supplementary files and I will leave this option to the journal’s administration to avoid any conflicts. Personally, I like to keep it as it’s.

Finally, this work should be much more concisely presented.  I would therefore recommend publication in Molecules only after minor revision.

  • I hope it is ok now.

Reviewer 2 Report

The manuscript is clearly written and of scientific merit. The compounds are well-characterized and the bioassays are scientifically sound. The language of the article is satisfactory, yet some minor improvements are still needed from the formal point of view.

The text should be spell-checked, as there are several spelling errors to be found. First of all, I would recommend to change the words "act as..." to "acting as..." in the title of the article.

The chemical name of the compound L2 in the experimental part (page 2 line 78) is not correct.

Throughout the manuscript the misspelled word "piperzine" is used, probably instead of the correct "piperazine".

Page 1 line 37: the words antitumor and anticancer are redundant, as they mean more or less the same, at least in common usage.

Page 4 line 147: "prepared by condensation of equimolar amounts" - this seems to be wrong, since the description in the experimental part indicates a 1:1.2 ratio between the starting substances. Please change accordingly.

Figure 2 has "intesity" instead of "intensity" in the upper part.

There are some minor errors also in the References part, e.g. in some cases abbreviations are used, in others not. Please check.

Author Response

Thanks for your review, Your comments enriched the manuscript.

The manuscript is clearly written and of scientific merit. The compounds are well-characterized and the bioassays are scientifically sound. The language of the article is satisfactory, yet some minor improvements are still needed from the formal point of view. The text should be spell-checked, as there are several spelling errors to be found. First of all, I would recommend to change the words "act as..." to "acting as..." in the title of the article.

  • Done

The chemical name of the compound L2 in the experimental part (page 2 line 78) is not correct.

  • Corrected to 2-(piperazin-1-yl)-N-(thiophen-2-ylmethylene)ethanamine (L2)

Throughout the manuscript the misspelled word "piperzine" is used, probably instead of the correct "piperazine".

  • Corrected in the whole MS.

Page 1 line 37: the words antitumor and anticancer are redundant, as they mean more or less the same, at least in common usage.

  • Have been standardized to antitumor 

Page 4 line 147: "prepared by condensation of equimolar amounts" - this seems to be wrong, since the description in the experimental part indicates a 1:1.2 ratio between the starting substances. Please change accordingly.

  • Corrected accordingly

Figure 2 has "intesity" instead of "intensity" in the upper part.

  • Corrected accordingly

There are some minor errors also in the References part, e.g. in some cases abbreviations are used, in others not. Please check.

  • Corrected accordingly

Reviewer 3 Report

  • Title: molecular docking instead “dock”
  • Spectras (Figures 1-4) should be transferred to the Supplementary files
  • abbreviation MEP should be explained at the first mention.
  • line 129: concentrations
  • Figure 6: Electrostatic potential around the molecule should be transparent in order to recognize the skeleton of molecules. The caption of Figure 6 should be more clear and colors of potential should be explained.
  • The molecular docking procedure was not described in Material and methods.
  • Page 14, molecular docking results: deeper discussion and comparing with published studies about pancreatic porcine lipase binding site and mode of interaction? It is not clear how did authors choose the binding site. Figures 11 are unclear; the text inside of figures is small and invisible. Figures have low resolution, also. The authors did not give the energy of interactions. Following those, molecular docking is not valid and should be removed from the manuscript.
  • Authors should be discussed about the relation between the antioxidant and antibacterial activity?

Author Response

Thanks for your review, your comments enriched the manuscript.

Title: molecular docking instead “dock”

  • Corrected

Spectras (Figures 1-4) should be transferred to the Supplementary files

  • I do not mind transferring or deleting any of the spectra figures to Supplementary files and I will leave this option to the journal’s administration to avoid any conflicts. Personally, I like to keep it as it’s.

abbreviation MEP should be explained at the first mention.

  • Corrected and inserted to the abstract as Molecular Electrostatic Potential (MEP),

line 129: concentrations

  • Corrected

Figure 6: Electrostatic potential around the molecule should be transparent in order to recognize the skeleton of molecules.

  • Done

The caption of Figure 6 should be more clear and colors of potential should be explained.

  • Done

The molecular docking procedure was not described in Material and methods.

  • Done Sir

Page 14, molecular docking results: deeper discussion and comparing with published studies about pancreatic porcine lipase binding site and mode of interaction? It is not clear how did authors choose the binding site.

  • The best bonding site for each ligand was chosen based on the lowest binding energy and RMSD
  • Discussed deeply

Figures 11 are unclear; the text inside of figures is small and invisible. Figures have low resolution, also. The authors did not give the energy of interactions. Following those, molecular docking is not valid and should be removed from the manuscript.

  • Binding energy values of each ligand with lipase was inserted into the text.
  • The resolution of the docking pictures was enhanced up to the page limits.
  • After that, if you find it not good, we do not mind to remove Figure 11 totally from the MS.

Authors should be discussed about the relation between the antioxidant and antibacterial activity?

  • In general, high antioxidant materials should be with high antibacterial activities. No clear relations between the antioxidant and antibacterial behaviors of the three desired Schiff base ligands were recorded. For example, L1 showed the highest antioxidant but a moderate antibacterial behaver.

Round 2

Reviewer 3 Report

The author authors have completely corrected the manuscript as recommended by the reviewer.

However, I think that Figure 11 does not satisfy the publishing quality and if the authors cannot post higher quality figures, my recommendation is to remove from the manusript.

Author Response

 Dear Sir

Please note that Figure 11 was removed totally from the manuscript as seen in the attached revised version file.

thank you 
